# Computer-Assisted Screening for Cervical Cancer Using Digital Image Processing of Pap Smear Images

**Kyi Pyar Win [1,\*], Yuttana Kitjaidure [1], Kazuhiko Hamamoto [2] and Thet Myo Aung [3]**

[1] Faculty of Engineering, King Mongkut's Institute of Technology Ladkrabang, Bangkok 10520, Thailand; yuttana.ki@kmitl.ac.th

[2] School of Information and Telecommunication Engineering, Tokai University, Tokyo 108-8619, Japan; hama@keyaki.cc.u-tokai.ac.jp

[3] Department of Pathology, Faculty of Medicine, Kale Hospital, Kale 02301, Myanmar; thetmyoaung29061991@gmail.com

\* Correspondence: 60601169@kmitl.ac.th or kyipyarwin5@gmail.com; Tel.: +66-64-473-7409

**Abstract:** Cervical cancer can be prevented by having regular screenings to find any precancers and treat them. The Pap test looks for any abnormal or precancerous changes in the cells on the cervix. However, the manual screening of Pap smear in the microscope is subjective with poorly reproducible criteria. Therefore, the aim of this study was to develop a computer-assisted screening system for cervical cancer using digital image processing of Pap smear images. The analysis of Pap smear image is important in the cervical cancer screening system. There were four basic steps in our cervical cancer screening system. In cell segmentation, nuclei were detected using a shape-based iterative method, and the overlapping cytoplasm was separated using a marker-control watershed approach. In the features extraction step, three important features were extracted from the regions of segmented nuclei and cytoplasm. RF (random forest) algorithm was used as a feature selection method. In the classification stage, bagging ensemble classifier, which combined the results of five classifiers—LD (linear discriminant), SVM (support vector machine), KNN (k-nearest neighbor), boosted trees, and bagged trees—was applied. SIPaKMeD and Herlev datasets were used to prove the effectiveness of our proposed system. According to the experimental results, 98.27% accuracy in two-class classification and 94.09% accuracy in five-class classification was achieved using the SIPaKMeD dataset. When the results were compared with five classifiers, our proposed method was significantly better in two-class and five-class problems.

**Keywords:** cervical cancer; Pap smear; watershed transform; random forest; ensemble classifier

## 1. Introduction

Cancer is the uncontrolled growth of abnormal cells in the body. Rather than responding appropriately to the signals that control normal cell behavior, cancer cells grow and divide in an uncontrolled manner, invading normal tissues and organs and eventually spreading throughout the body [1]. Cervical cancer is cancer arising from the cervix. The most important risk factor for cervical cancer is infection with human papillomavirus (HPV). The goal of cervical screening is to identify and remove significant precancerous lesions in addition to preventing mortality from invasive cancer. Cervical cancer is the fourth most frequent cancer in women, with an estimated 570,000 new cases in 2018, representing 6.6% of all female cancers. Approximately 90% of deaths from cervical cancer occur in low- and middle-income countries. Precancerous changes in the cervix usually don't cause any signs or symptoms [2]. Symptoms of cervical cancer include irregular intermenstrual (between periods) or abnormal vaginal bleeding after sexual intercourse, back, leg or pelvic pain, fatigue, weight

loss, loss of appetite, vaginal discomfort or odorous discharge, and a single swollen leg. More severe symptoms may arise at advanced stages [3].

Cervical cancer can be prevented by regular screening tests if precancerous changes are detected and treated effectively before cancer develops. Cervical cancer typically develops from precancerous changes over 10 to 20 years. The only way to know if there are abnormal cells in the cervix, which may develop into cervical cancer, is to have a cervical screening test. Screening is testing of all women at risk of cervical cancer, most of whom will be without symptoms. A Pap test is commonly used to screen for cervical cancer. A Pap smear is a simple, quick, and essentially painless screening test (procedure) for cancer or precancer of the uterine cervix. Cervical cancer testing should start at age 21. Women under age 21 should not be tested. Women between the ages of 21 and 65 should have a Pap test done every 3 years. About 80 percent of deaths from cervical cancer occur in developing countries due to the lack of screening programs [4].

The regular Pap test system reduces the incidence rate of cervical cancer. The visual examination of the Pap smears is time-consuming, very demanding, tedious, and expensive in terms of labor requirements. The cytotechnologists are laboratory professionals who study cells and cellular anomalies who go through specialized training, typically of about one year. The Pap test or smear starts with the pelvic exam. In this exam, cell samples are collected from the cervix and stained on a glass slide. The  collected cells are visually examined under a microscope to classify each cell. The shape, size, texture, and nucleus to cytoplasm ratio are the important features to classify the cervical cells into normal and abnormal epithelial cells [5].

Literature reviews that are related to previous research studies about cervical cancer cells' segmentation and classification are described as follows: A. H. Mbaga et al. proposed Pap smear images classification for cervical cancer detection using a support vector machine (SVM) classifier and got an accuracy of 92.961% [6]. In another study, M. E. Plissiti et al. presented an approach to segment cells cluster using intensity variation. But most of the real Pap smear images were poor contrast, and sometimes intensity variation was invisible in overlapping conditions [7]. S. N. Sulaiman et al. also developed the method of overlapping cell separation by integrating edge detection and pseudo-color algorithm. The seed-based region growing method was used to detect the boundary of cells, and cell components (nucleus, cytoplasm, and background) were grouped according to color techniques [8]. Most research studies focus on the nuclei segmentation in a single [9] or overlapping cells [10], and other researchers focus on the nuclei segmentation in both single and overlapping cells [11].

The studies of computer-assisted screening of cervical cytology can be classified into the cell level classification, smear level classification, applied segmentation algorithms, usage of features, and classifiers. In cell level classification, each input image has only one cell, which is classified into the normal or abnormal cell [12]. In smear level classification, the input image includes one more cell and also other artifacts [13]. There are many different algorithms that have been used for image segmentation, such as clustering [14], thresholding [15,16], edge detection [17], and watershed transformation [18]. The watershed segmentation was firstly proposed by L. Vincent and P. Soille [19]. Over segmentation is one of the significant problems in watershed algorithms. Our proposed study developed an improved marker-based watershed algorithm that provided better results than the traditional algorithm and helped to reduce the over-segmentation problem. Most research studies have used a watershed algorithm to split overlapping cells' nuclei [20]. But in our multi-cells Pap smear images, overlapping cytoplasm was mostly contained, as shown in Figure 1 and a few overlapping nuclei were found. Therefore, the watershed transform algorithm was used for overlapping cytoplasm segmentation.

Recently, Chuanyun et al. [21] had proposed the segmentation of nuclei and cytoplasm regions using the gradient vector flow method (GVF). However, this study only used a single cell as an input to be analyzed. While each cell slide can contain over 10,000 cells [22], the approach using a single cell cannot be enough in real cases. However, it does not solve the issue of the overlapping cells. Many different algorithms have been proposed to solve the problem of overlapping cells [23]. Among them, marker controlled watershed transformation is one of the most used methods. The

main problem in this approach is over-segmentation problems [24]. In feature extraction, mostly used features are shape [25], texture [26], and color features. The major advantage of using texture attribute is its simplicity. Therefore, the texture feature was extracted using GLCM (Gray Level Co-Occurrence Matrix) in our feature extraction stage. The mostly used classifiers in the multi-cell cervical image analysis are support vector machine (SVM) [27], LDA (Linear Discriminant Analysis) [28], k-nearest neighbor (KNN) [29], and ANN (Artificial Neural Networks) [30]. There have been many research studies about cervical cancer detection, but most studies have only targeted the segmentation of nuclei regions [31]. The segmentation of cytoplasm regions is also essential. The features that are extracted from cytoplasm regions are helpful for the classification of abnormal cells [32].

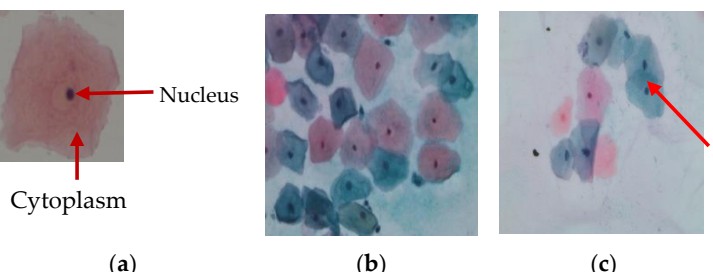

(a)  (b)  (c)

**Figure 1.** (**a**) Single-cell image (nucleus and cytoplasm), (**b**) Multi-cells image, and (**c**) Overlapping cytoplasm.

This paper is divided into four parts. Part 1 introduces about cervical cancer screening system and discusses the previous research studies. Part 2 expresses the explanations of the datasets used and the methodology that is used in this study. Part 3 indicates the results of the proposed system. Finally, Part 4 presents the conclusion.

## 2. Materials and Methods

The system flow diagram of the proposed computer-assisted screening of cervical cytology is presented in Figure 2. Our proposed system involved six stages—Image acquisition, image enhancement, cell segmentation, features extraction, features selection, and classification [33]. At the image acquisition step, the SIPaKMeD dataset was used for multi-cells, and the Herlev dataset was used for a single cell. Input Pap smear images were enhanced and denoised to improve the image quality as an image enhancement step. The next step was cell segmentation. This step partitioned the input images into the interesting regions—nucleus and cytoplasm. After segmentation, the next step was feature extraction. In feature extraction, distinctive interested points or features were extracted. In the features selection stage, the random forest algorithm was used as a selection method. The final step was the classification. In this stage, the cells were classified using bagging ensemble classifiers into normal or abnormal cells. The details of each step have been explained in each sub-sections.

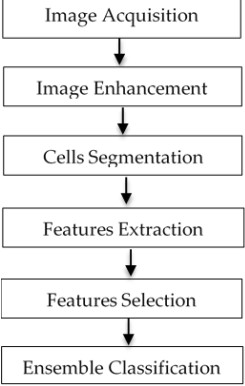

**Figure 2.** The methodology of proposed cervical cancer detection and classification system.

*2.1. Image Acquisition*

For image acquisition, we used two datasets named SIPaKMeD [34] and Herlev datasets [35]. For single-cell classification, the Herlev dataset was used, and the SIPaKMeD dataset was used for multi-cells classification. The Herlev dataset contained 917 images. The classes 1 to 3 are normal cervical cells, and classes 4 to 7 are abnormal cervical cells. In multi-cells dataset, there were 966 images, and 4049 cells were cropped from these images. Cells were divided into normal, begin, and abnormal stage. There were five classes—superficial intermediate cells, parabasal cells, metaplastic cells, dyskeratotic cells, and koilocytotic cells. The details of each dataset have been explained in Tables 1 and 2. The sample pap smear images of Herlev dataset and SIPaKMeD dataset were shown in Figures 3 and 4.

**Table 1.** Descriptions of seven-classes cells from the Herlev (single cells) dataset.

| Class | Number of Cells |
|---|---|
| Normal Cells | |
| 1. Normal superficial cells | 74 |
| 2. Normal intermediate cells | 70 |
| 3. Normal columnar cells | 98 |
| Abnormal Cells | |
| 4. Mild dysplastic cells | 182 |
| 5. Moderate dysplastic cells | 146 |
| 6. Severe dysplastic cells | 197 |
| 7. Carcinoma in situ | 150 |
| Total | 917 |

**Table 2.** Descriptions of five-classes cells from the SIPaKMeD (Multi-cells) dataset.

| Class | Number of Images | Number of Cells |
|---|---|---|
| Normal Cells | | |
| 1. Superficial-Intermediate cells | 126 | 831 |
| 2. Parabasal cells | 108 | 787 |
| Benign Cells | | |
| 3. Metaplastic cells | 271 | 793 |
| Abnormal Cells | | |
| 4. Dyskeratotic cells | 223 | 813 |
| 5. Koilocytotic cells | 238 | 825 |
| Total | 966 | 4049 |

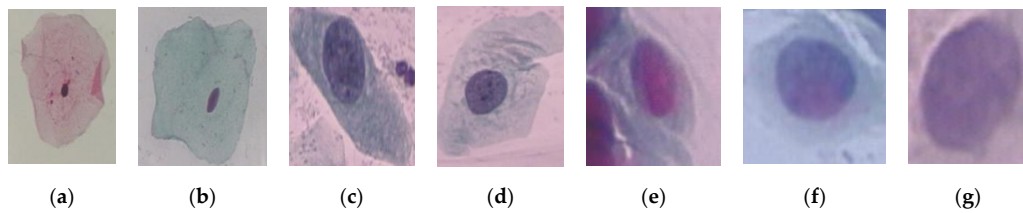

|     (a)     |     (b)     |     (c)     |     (d)     |     (e)     |     (f)     |     (g)     |

**Figure 3.** Single cells images of seven classes: (**a**) Superficial squamous epithelia, (**b**) Intermediate squamous epithelia, (**c**) Columnar epithelial, (**d**) Mild squamous non-keratinizing dysplasia, (**e**) Moderate squamous non-keratinizing dysplasia, (**f**) Severe squamous non-keratinizing dysplasia, (**g**) Squamous cell carcinoma in situ.

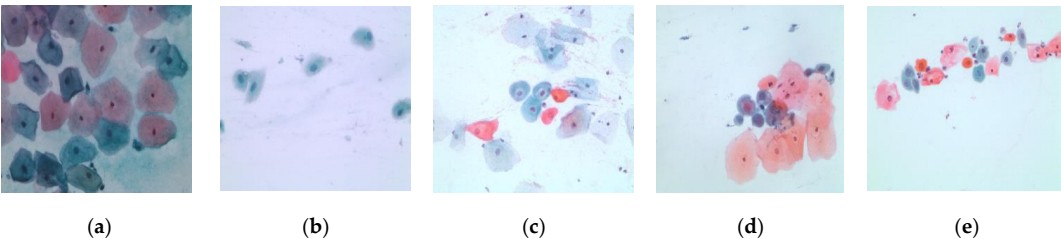

**Figure 4.** Multi-cells images of five classes: (**a**) Superficial-Intermediate cells, (**b**) Parabasal cells, (**c**) Metaplastic cells, (**d**) Dyskeratotic cells, and (**e**) Koilocytotic cells.

## 2.2. Image Enhancement

Most of Pap smear images were noisy and low contrast, as shown in Figure 5a. Therefore, image enhancement was needed to remove the noises and increase the contrast. A median filter was used to remove the noises, as shown in Figure 5b, and CLAHE (contrast limited adaptive histogram equalization) was used to enhance the contrast, as shown in Figure 5c. High contrast images were easier and more precise for cell segmentation stage than in low contrast images.

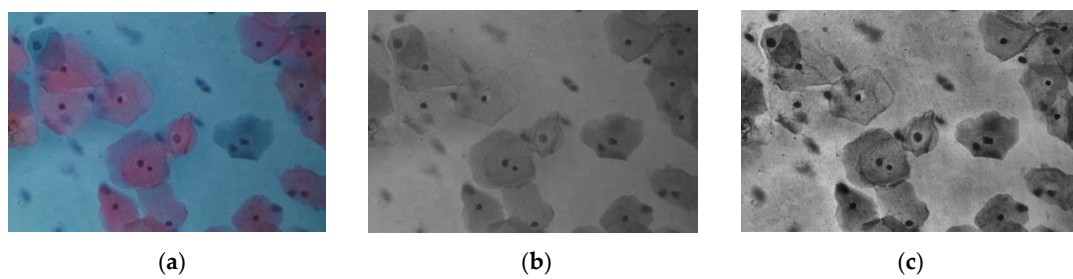

**Figure 5.** Image enhancement, (**a**) original image, (**b**) noise removal by the medial filter, and (**c**) contrast enhancement by CLAHE (contrast limited adaptive histogram equalization).

## 2.3. Cells Segmentation

The aim of this step was to segment the regions of the cell from input images. The nuclei and cytoplasm are important components in the cell region. In a Pap smear screening system, cytologists examine the microscope images of cells and label the cells into cancer or normal cells based on the appearance of cells components. The automated screening system is also the same procedure. The segmentation of cell components is an important step in the automated detection system. There are many difficulties in the multi-cells segmentation process, such as overlapping cells or including unwanted artifacts. The nuclei segmentation is easier than cytoplasm segmentation. In our multi-cells image, nuclei were low intensity, and the shapes were well structured, mostly oval or round shape, and significantly different from the other regions, background, or cytoplasm. But the major issues in cytoplasm segmentation are overlapping boundary and poor contrast. Most studies have focused on nuclei segmentation, and rarely studies have focused on cytoplasm segmentation. In Pap smear analysis, the characteristics of cytoplasm are very important. We used the marker-controlled watershed algorithm for cytoplasm segmentation to solve the issue of overlapping boundary detection and splitting of touching cells into individual cells. The main problem of the standard watershed transform is over-segmentation. To avoid this problem, we used markers. The marker image was a binary image with one marker point or multiple points. The flowchart of the proposed modified watershed transform algorithm is described in Figure 6.

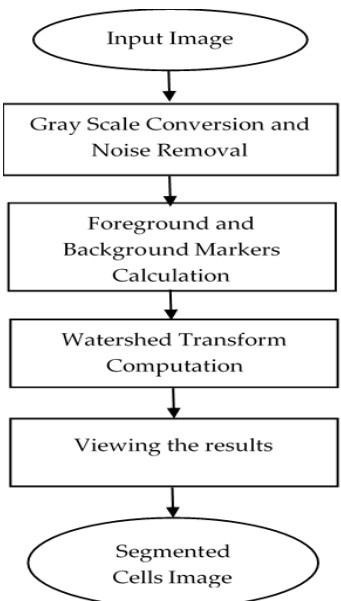

**Figure 6.** Flowchart of proposed marker controlled watershed transformation.

In our proposed overlapping cells' segmentation method, there were ten stages to segment the multi-cells images into individual cells that were used for the nuclei and cytoplasm regions extraction. The summary of each step is shown in Table 3. As first, the original images were changed into gray images. The next step was foreground and background markers extraction and then segmentation using watershed transform function and viewing the segmented results. For foreground and background markers calculation, morphological operations based techniques were used. To separate overlapping cells into individual cells, the boundary of cytoplasm regions was detected after overlapping the cells' detection stage. Then, the area of each cell was detected by thresholding the predefined minimum and maximum area values to remove the unwanted object areas. After that, the detected regions were cropped by a bounding box, and the cropped regions were classified into three classes using unsupervised machine learning, k-means with six intensity-based features. The intensity variation of three groups of cell patches (isolated, touching, and overlapping cells) were significantly different, as shown in Figure 7. So, we used six intensity-based features (mean, variance, skewness, kurtosis, energy, and entropy).

**Table 3.** Processing steps of proposed overlapping cells' segmentation method.

**Step 1:** Read color image and convert gray image
**Step 2:** Mark the foreground objects
**Step 3:** Compute background objects
**Step 4:** Use markers' image that is roughly in the middle of the cells to be segmented
**Step 5:** Compute the watershed transform of makers' image
**Step 6:** Show the result of detected overlapping cells' regions
**Step 7:** Calculate the boundaries of detected regions in the image
**Step 8:** Detect areas between the minimum and maximum values for cells regions
**Step 9:** Cropping the regions
**Step 10:** Classify the regions of the cell into isolated, touching, or overlapped cells

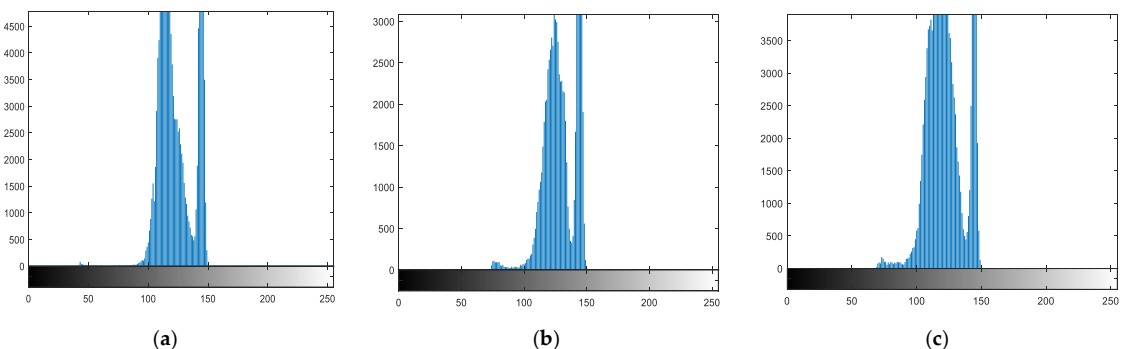

**Figure 7.** Histograms of (**a**) isolated cell, (**b**) touching cell, and (**c**) overlapped cell.

K-means clustering approach was used to classify the cropping cells' patches into three groups. It was first proposed by McQueen [36]. It organizes a set of observations that are represented as feature vectors into clusters based on their similarity. There are three basic steps in the training algorithm for k-means. They are initialization, update, and assignment. Initialization assigns each observation from the data set randomly to one of the k clusters and then takes k observations randomly from the data set and assigns each to a cluster. Figure 8 shows the three steps of the k-means clustering algorithm for classification. Results of the proposed cell segmentation algorithm are showed in Figure 9.

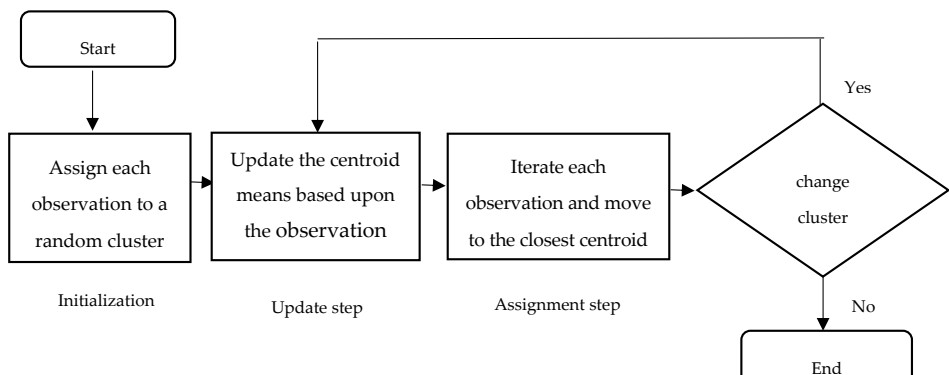

**Figure 8.** K-means clustering algorithm.

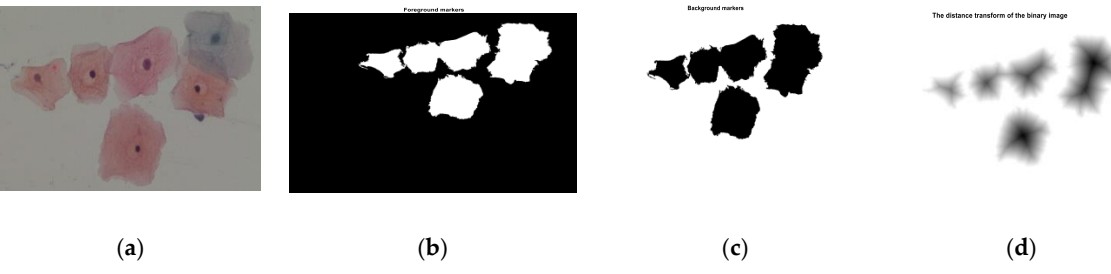

**Figure 9.** *Cont.*

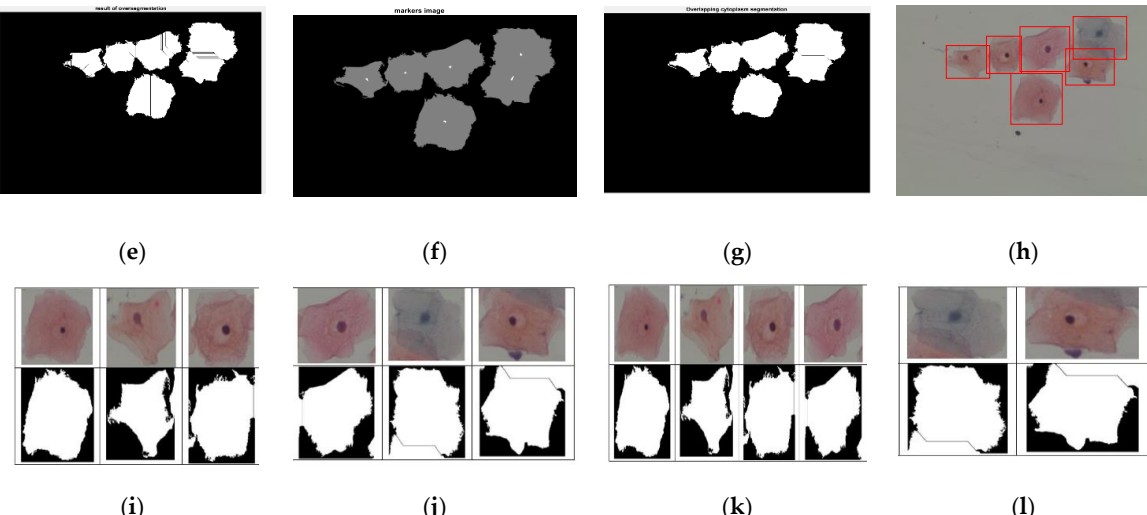

**Figure 9.** Results of the proposed cell segmentation algorithm. (**a**) Original color image, (**b**) Image of foreground objects, (**c**) Image of background objects, (**d**) Image of distance transformed, (**e**) Image of over-segmentation result without using markers, (**f**) Markers' image, (**g**) Image of segmentation result by using markers' image, (**h**) Image of boundary detection, (**i,j**) Image of cropping results, (**k**) Image of isolated or touching cells, and (**l**) Image of overlapping cells.

## 2.4. Nuclei and Cytoplasm Segmentation

In this stage, nuclei and cytoplasm regions from each segmented cell that resulted from overlapping cells' segmentation stage were segmented. There were three types of segmented cells that were outputs of overlapping cells' segmentation stage. They were isolated cells, touching cells, and overlapping cells. We divided the nuclei and cytoplasm segmentation into three sub-processes. The first one was the segmentation of the components of the cell from isolated cells. The second one was segmentation from touching cells, and the last one was segmentation from overlapping cells. The cytoplasm boundary of isolated cells, touching cells, and overlapping cells could be extracted from segmented cells' results of the watershed transform approach that was proposed in the overlapping cells' segmentation stage. The regions of touching and overlapping cytoplasm in the image obtained in the segmentation step were not enough to represent the boundaries of the cytoplasm. Thus, we did the process of smoothing the boundaries of the cytoplasm using an edge smoothing method that is described in Table 4.

**Table 4.** Processing steps of edges smoothing method.

Step 1: Read grayscale image and convert binary image
Step 2: Extract the largest blob only
Step 3: Crop-off the frame on the left and top
Step 4: Fill holes
Step 5: Blur the image
Step 6: Threshold again
Step 7: Show the smoothed binary image

The nuclei of isolated cells, touching cells, and overlapping cells were segmented using the shape-based iteration method that is described in Table 5.

**Table 5.** Processing steps of the shape-based nuclei detection algorithm.

Step 1: Read the input color image and invert the grayscale image
Step 2: Remove noise using the median filter
Step 3: Predefine minimum area, major and minor axis lengths, minimum and maximum intensity values, and solidity
Step 4: Binarize the image using the lowest and highest thresholds
Step 5: Remove the regions under limited shape and intensity values
Step 6: Segment nuclei

The nuclei segmentation was easier than cytoplasm segmentation. In our multi-cells image, nuclei were low intensity, and the shapes were well structured, mostly oval or round shape, and significantly different from the other regions, background, or cytoplasm. The area, average intensity, major and minor axis lengths, and solidity were the five most important features of nuclei to distinguish from other objects, cytoplasm, and background. Therefore, we used these features and developed a method to detect and segment nuclei. A median filter was used to remove the noise from the original Pap smear images. The shape-based features (area, intensity values, major axis length, minor axis length, and solidity) were used in the nuclei segmentation process. According to the experiments, the minimum area of nuclei was 362 μm$^2$. The average intensity value was between 60 and 150, and solidity values were less than or equal 0.98. The major axis length of nuclei was between 24 m and 117 m, and the minor axis length was between 17m and 87m. Shape-based features' name and their formula are explained in Figure 10b.

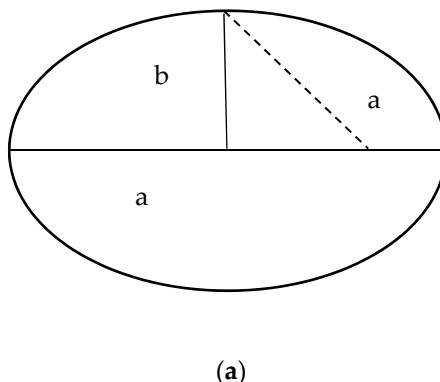

| Feature | Formula |
|---|---|
| Area | $\pi a b$ |
| Intensity values | Mean value |
| Major axis length | $2 * a$ |
| Minor axis length | $2 * b$ |
| Solidity | $\dfrac{Area}{Convex\ area}$ |

(**a**)                                                                                  (**b**)

**Figure 10.** (**a**) Sample diagram of calculation formula for extracted features. (**b**) Shape-based features and their equations.

The results of the nuclei segmentation step are shown in Figures 11 and 12. The performance of our proposed nuclei detection method was evaluated, and their results are shown in Table 6. The nuclei and cytoplasm segmentation results of isolated cells, touching cells, and overlapping cells are described in Tables 7–9, respectively. The proposed overlapping cells' segmentation method had been tested on images with a 4049 cytoplasm in total. About 6.47% of the cytoplasm was isolating, while 35.27% of cytoplasm were touching, and 58.26 % of cytoplasm were overlapping. The rate of well-segmented cytoplasm was calculated by dividing the detected cytoplasm number with the actual cytoplasm number and multiplied by 100, and the results are shown in Table 10.

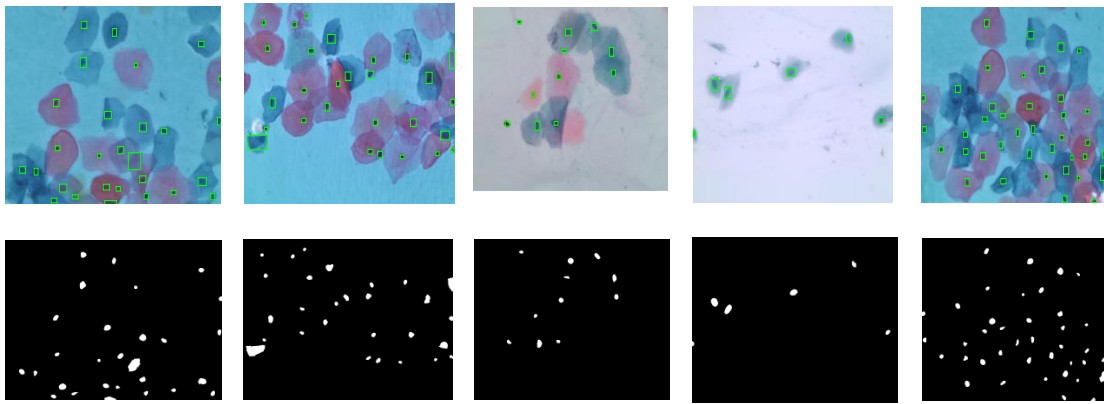

**Figure 11.** Nuclei detection in multi-cells Pap smear images.

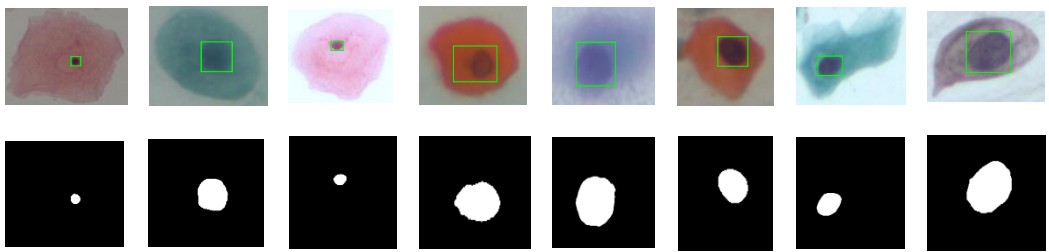

**Figure 12.** Nuclei detection from cropping single cells' patches.

**Table 6.** The performance of our proposed nuclei detection method.

| DSC | FNR | TPR |
|-----|-----|-----|
| 0.862 | 0.500 | 0.945 |

The dice similarity coefficient (DSC)—the value was between 0 and 1. False-negative rate (FNR)—the total number of pixels that were incorrectly classified as a nucleus. True positive rate (TPR)—the total number of pixels that were correctly classified as a nucleus.

**Table 7.** Nuclei and cytoplasm Segmentation for isolated cells.

| Class | Isolated Cell | Segmented Cytoplasm | Smooth Boundary | Segmented Nuclei |
|-------|---------------|---------------------|-----------------|------------------|
| Class1 | | | | |
| Class2 | | | | |

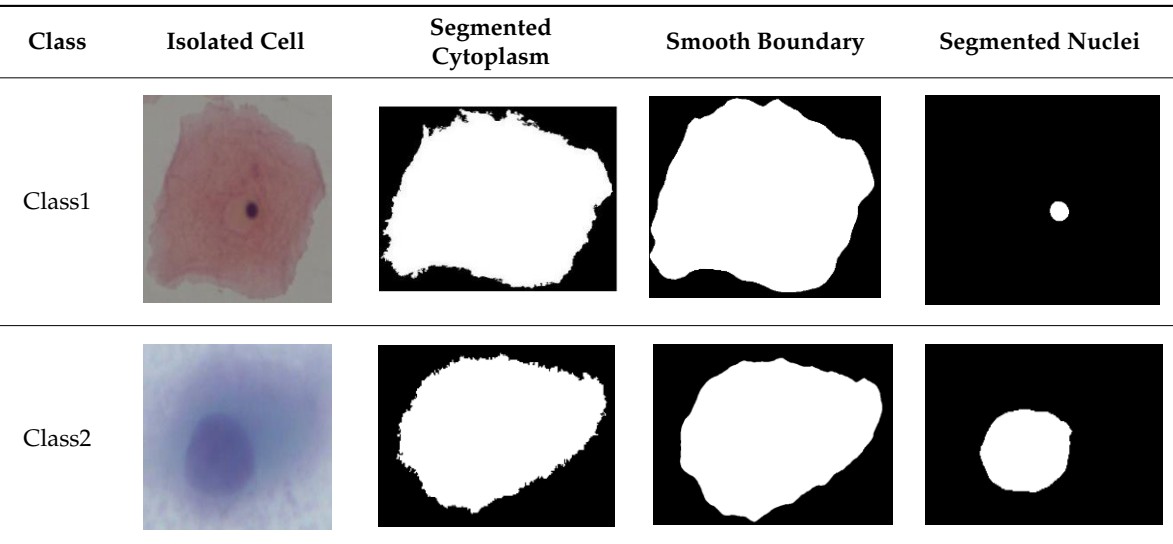

**Table 7.** *Cont.*

| Class | Isolated Cell | Segmented Cytoplasm | Smooth Boundary | Segmented Nuclei |
|---|---|---|---|---|
| Class3 |  |  |  |  |
| Class4 |  |  |  |  |
| Class5 |  |  |  |  |

**Table 8.** Nuclei and Cytoplasm Segmentation for touching cells.

| Class | Touching Cell | Segmented Cytoplasm | Clean Boundary | Smooth Boundary | Segmented Nuclei |
|---|---|---|---|---|---|
| Class1 |  |  |  |  |  |
| Class2 |  |  |  |  |  |
| Class3 |  |  |  |  |  |
| Class4 |  |  |  |  |  |

**Table 8.** *Cont.*

| Class | Touching Cell | Segmented Cytoplasm | Clean Boundary | Smooth Boundary | Segmented Nuclei |
|-------|---------------|---------------------|----------------|-----------------|------------------|
| Class5 |  |  |  |  |  |

**Table 9.** Nuclei and Cytoplasm Segmentation for overlapping cells.

| Class | Overlapping Cell | Segmented Cytoplasm | Clean Boundary | Smooth Boundary | Segmented Nuclei |
|-------|------------------|---------------------|----------------|-----------------|------------------|
| Class1 |  |  |  |  |  |
| Class2 |  |  |  |  |  |
| Class3 |  |  |  |  |  |
| Class4 |  |  |  |  |  |
| Class5 |  |  |  |  |  |

**Table 10.** The rate of well-segmented cytoplasm in terms of the number of cytoplasms.

| Isolated Cytoplasm | Touching Cytoplasm | Overlapping Cytoplasm | Total |
|--------------------|--------------------|-----------------------|-------|
| 262 | 1428 | 2359 | 4049 |
| 100% | 95.85% | 77.59% | 95.94% |

## 2.5. Features Extraction

After the cells' segmentation stage, the next stage was features extraction. In this stage, the important features, texture, shape, and color features were extracted. Texture features were obtained using the gray level co-occurrence matrix. In the cervical Pap smear image, healthy and abnormal cells were highly different in their distributions of color and shape [37]. Therefore, we extracted features of shape and color. The extracted texture features were contrast, smoothness, third moment, uniformity, and entropy for RGB channels [38]. The color-based features were mean of six color channels (red, green, and blue) and H, S, and V channels (hue, saturation, and value) that were extracted independently from RGB and HSV model. The extracted features' names of nuclei and cytoplasm are shown in Table 11.

**Table 11.** The descriptions of nuclei and cytoplasm features' names.

| No. | Nuclei Features (35) | No. | Cytoplasm Features (35) |
|-----|----------------------|-----|-------------------------|
| N1 | Nucleus's area | C1 | Cytoplasm's area |
| N2 | Nucleus's major axis length | C2 | Cytoplasm's major axis length |
| N3 | Nucleus's minor axis length | C3 | Cytoplasm's minor axis length |
| N4 | Nucleus's eccentricity | C4 | Cytoplasm's eccentricity |
| N5 | Nucleus's orientation | C5 | Cytoplasm's orientation |
| N6 | Nucleus's equivalent diameter | C6 | Cytoplasm's equivalent diameter |
| N7 | Nucleus's solidity | C7 | Cytoplasm's solidity |
| N8 | Nucleus's extent | C8 | Cytoplasm's extent |
| N9 | Nucleus's compactness | C9 | Cytoplasm's compactness |
| N10 | Nucleus's short diameter | C10 | Cytoplasm's short diameter |
| N11 | Nucleus's long diameter | C11 | Cytoplasm's long diameter |
| N12 | Nucleus's elongation | C12 | Cytoplasm's elongation |
| N13 | Nucleus's roundness | C13 | Cytoplasm's roundness |
| N14 | Nucleus's perimeter | C14 | Cytoplasm's perimeter |
| N15 | Nucleus's position | C15 | Nucleus to cytoplasm ratio |
| N16 | Nucleus's maximum number | C16 | Cytoplasm's maximum number |
| N17 | Nucleus's minimum number | C17 | Cytoplasm's minimum number |
| N18 | Nucleus's average intensity in R | C18 | Cytoplasm's average intensity in R |
| N19 | Nucleus's average intensity in G | C19 | Cytoplasm's average intensity in G |
| N20 | Nucleus's average intensity in B | C20 | Cytoplasm's average intensity in B |
| N21 | Nucleus's average intensity in H | C21 | Cytoplasm's third moment in H |
| N22 | Nucleus's average intensity in S | C22 | Cytoplasm's uniformity in S |
| N23 | Nucleus's average intensity in V | C23 | Cytoplasm's entropy in V |
| N24 | Nucleus's contrast | C24 | Cytoplasm's contrast |
| N25 | Nucleus's local homogeneity | C25 | Cytoplasm's local homogeneity |
| N26 | Nucleus's correlation | C26 | Cytoplasm's correlation |
| N27 | Nucleus's cluster shape | C27 | Cytoplasm's cluster shape |
| N28 | Nucleus's cluster prominence | C28 | Cytoplasm's cluster prominence |
| N29 | Nucleus's maximum probability | C29 | Cytoplasm's maximum probability |
| N30 | Nucleus's energy | C30 | Cytoplasm's energy |

**Table 11.** *Cont.*

| No. | Nuclei Features (35) | No. | Cytoplasm Features (35) |
|---|---|---|---|
| N31 | Nucleus's variance | C31 | Cytoplasm's variance |
| N32 | Nucleus's uniformity | C32 | Cytoplasm's uniformity |
| N33 | Nucleus's entropy | C33 | Cytoplasm's entropy |
| N34 | Nucleus's sum entropy | C34 | Cytoplasm's sum entropy |
| N35 | Nucleus's difference entropy | C35 | Cytoplasm's difference entropy |

### 2.6. Features Selection

In this stage, the random forest algorithm was used as a feature selection algorithm. The main reason for using a feature selection method was to select the most important features and to improve the accuracy of the classifier. The feature selection algorithm can reduce the complexity of the classification model and reduce the training time of machine learning algorithms. There are many feature selection algorithms. Among them, we used the random forest algorithm because the tree-based strategies used by random forests naturally rank by how well they improve the purity of the node. This means a decrease in impurity over all trees. Nodes with the greatest decrease in impurity were at the start of the trees, while nodes with the least decrease in impurity were at the end of trees. Thus, by pruning trees below a node, we could create a subset of the most important features. Table 12 shows the features' names and their top rank attributes by the random forest (RF) algorithm.

**Table 12.** Features' names and their top rank attributes by random forest (RF) algorithm.

| No. | Selected Features Name | Ranked Values |
|---|---|---|
| 1 | Nucleus to cytoplasm ratio | 0.67559 |
| 2 | Nucleus's average intensity in G | 0.58192 |
| 3 | Cytoplasm's average intensity in R | 0.56378 |
| 4 | Nucleus's average intensity in R | 0.5015 |
| 5 | Cytoplasm's average intensity in G | 0.48555 |
| 6 | Nucleus's entropy | 0.39472 |
| 7 | Nucleus's average intensity in B | 0.38415 |
| 8 | Nucleus's uniformity | 0.32821 |
| 9 | Cytoplasm's contrast | 0.27581 |
| 10 | Nucleus's long diameter | 0.25963 |
| 11 | Cytoplasm's average intensity in B | 0.24524 |
| 12 | Cytoplasm's long diameter | 0.23685 |
| 13 | Cytoplasm's uniformity | 0.23395 |
| 14 | Nucleus's perimeter | 0.21901 |
| 15 | Cytoplasm's major axis length | 0.19202 |
| 16 | Cytoplasm's equivalent diameter | 0.18936 |
| 17 | Nucleus's area | 0.17126 |
| 18 | Cytoplasm's perimeter | 0.16393 |
| 19 | Nucleus's minimum number | 0.16279 |
| 20 | Nucleus's minor axis length | 0.15295 |

### 2.7. Classification

In this stage, we used a bagging ensemble classifier. Ensemble learning can help to improve the prediction results by combining several models. Bagging uses bootstrap sampling to obtain the data subsets for training the base learners. For combining the outputs of base learners, bagging uses voting for classification. Combining stable learners was less advantageous since the ensemble would not help improve generalization performance. In our proposed classifier, five classifiers were trained, and the results of the predictions of each classifier were combined. The result was decided based on majority voting. The block diagram of the combined five classifiers is shown in Figure 13.

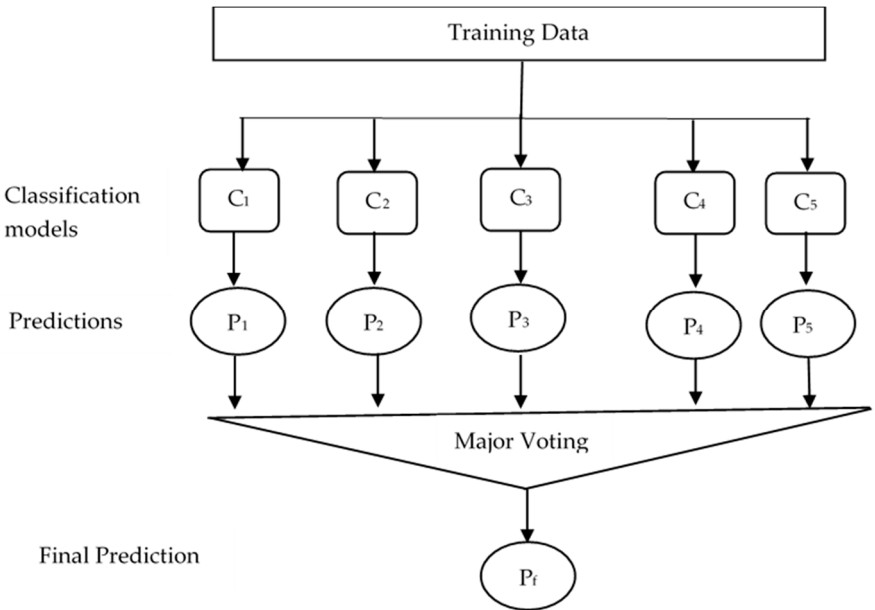

**Figure 13.** Block diagram of ensemble classifiers. (where C1 = LD, C2 = SVM, C3 = KNN, C4 = Boosted trees, and C5 = Bagged trees). LN, linear discriminant; SVM, support vector machine; KNN, k-nearest neighbor.

## 3. Results

To validate the effectiveness of our proposed system, SIPaKMeD (multi-cells) dataset and Herlev dataset (single-cell) were used. In the multi-cells dataset, there were 996 images, and 4049 cells were cropped from these total images. These cells were divided into five classes, class1 (superficial intermediate cells), class 2 (parabasal cells), class 3 (metaplastic cells), class 4 (dyskeratotic cells), and class 5 (koilocytotic cells). For the two-class problem, the first three classes were grouped into normal cells and 1618 cells in total. The last two classes were grouped into abnormal cells and 2431 cells in total. The performance measures were accuracy, recall, specificity, precision, and F-measure [39]. The formulas are given in the below equations:

$$\text{Accuracy} = \frac{TP + TN}{TP + FP + TN + FN} \tag{1}$$

$$\text{Sensitivity} \backslash \text{Recall} = \frac{TP}{TP + FN} \tag{2}$$

$$\text{specificity} = \frac{TN}{TN + FP} \tag{3}$$

$$\text{Precision} = \frac{TP}{TP + FP} \tag{4}$$

$$\text{F\_Measure} = 2 * \frac{\text{Precision} * \text{Recall}}{\text{Precision} + \text{Recall}} \tag{5}$$

where True positive (TP)—the results of the correctly classified positive class. True negative (TN)—the results of the correctly classified negative class. False-positive (FP)—the results of the incorrectly classified positive class. False-negative (FN)—the results of the incorrectly classified negative class.

The comparison results of classification performance in terms of accuracy with five classifiers and our ensemble classifier using nuclei features only, cytoplasm features only, combining nuclei and cytoplasm features without features selection method, and with features selection method for the two-class problem (SIPaKMed dataset) are shown in Figure 14 and five-class problem in Figure 15.

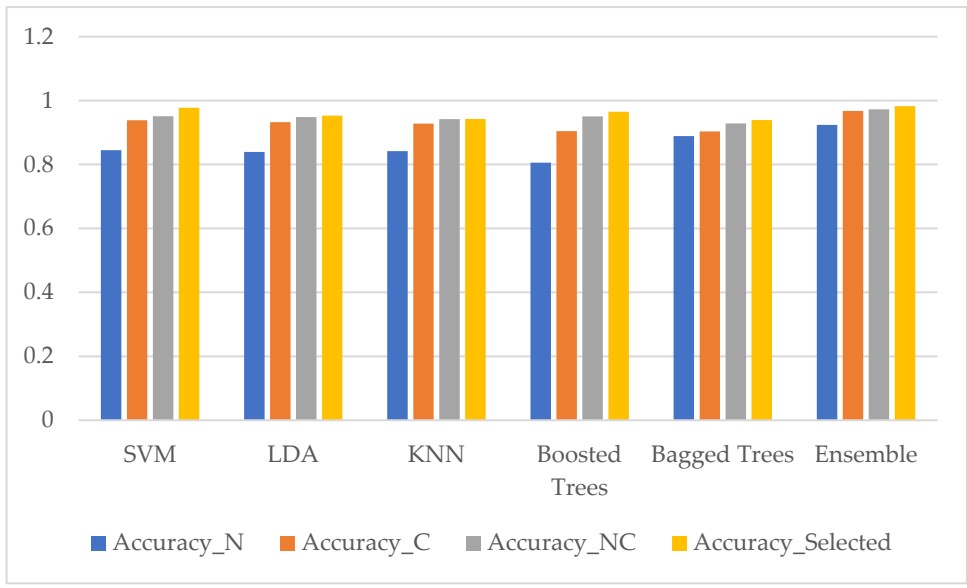

**Figure 14.** Classification performance of five classifiers and our ensemble classifier in terms of accuracy using four datasets (nuclei features only (Accuracy_N), cytoplasm features only (Accuracy_C), combining nuclei and cytoplasm features without features selection method (Accuracy_NC), and with features selection method (Accuracy_Selected)). (SIPaKMed dataset for the two-class problem).

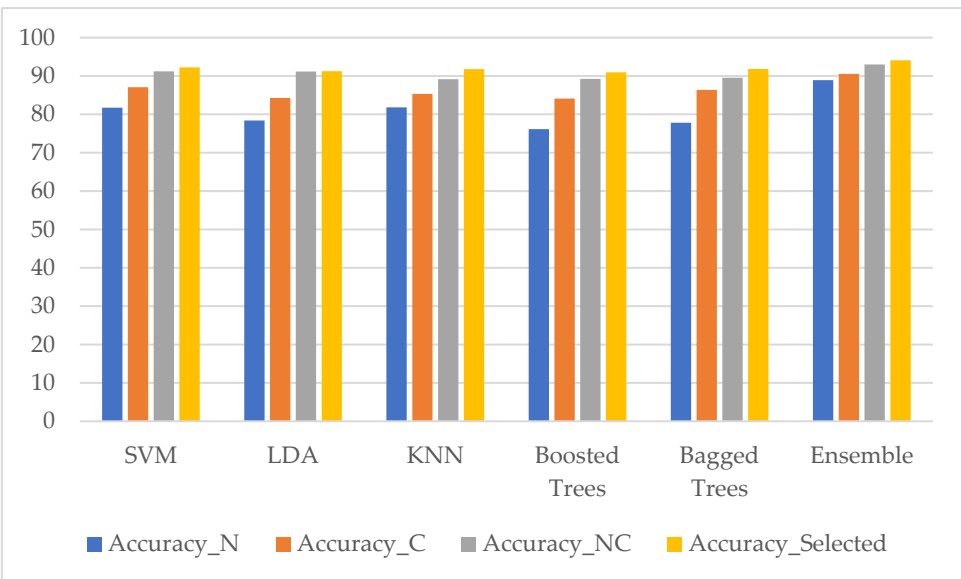

**Figure 15.** Classification performance of five classifiers and our ensemble classifier in terms of accuracy using four datasets (nuclei features only (Accuracy_N), cytoplasm features only (Accuracy_C), combining nuclei and cytoplasm features without features selection method (Accuracy_NC), and with features selection method (Accuracy_Selected)). (SIPaKMed dataset for the five-class problem).

Moreover, the five performance measures of each classifier using selected features are shown in Figure 16 for the two-class problem and Figure 17 for the five-class problem.

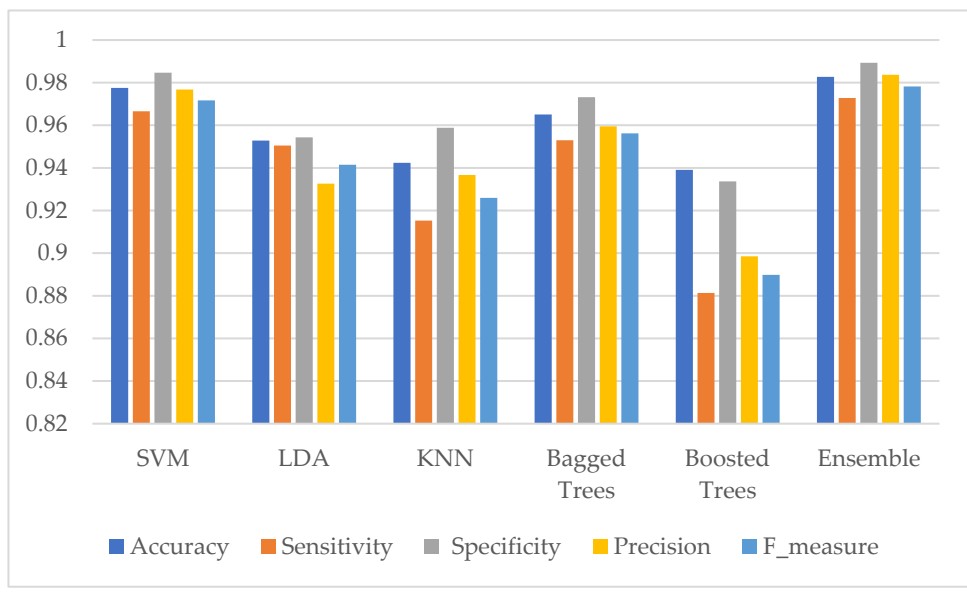

**Figure 16.** Evaluation of classifier's performance using selected features for the SIPAKMed dataset in the two-class problem.

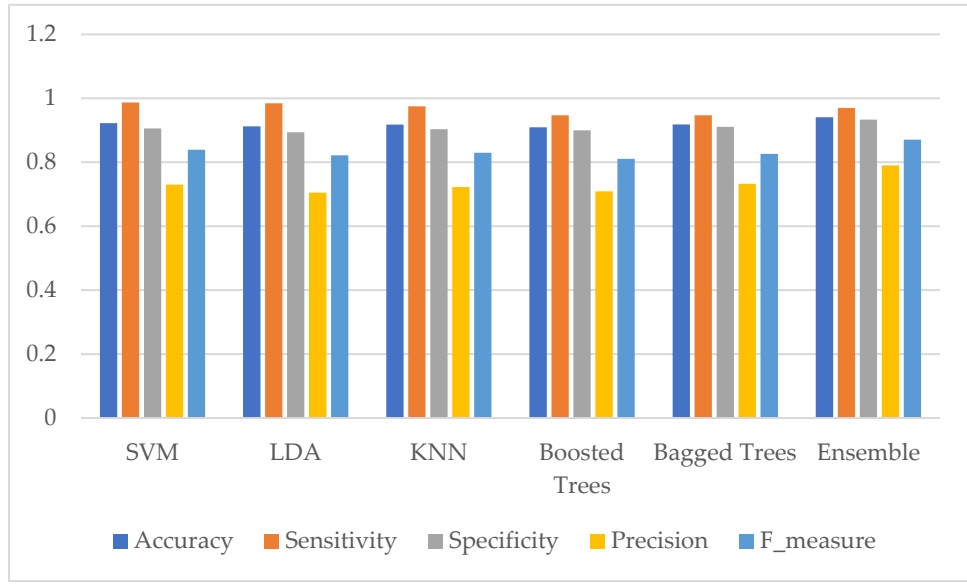

**Figure 17.** Evaluation of classifier's performance using selected features for the SIPAKMed dataset in the five-class problem.

The comparison results of five classifiers with our ensemble classifier in terms of accuracy using four datasets (nuclei features only, cytoplasm features only, combining nuclei and cytoplasm features without features selection method, and with features selection method) are shown in Figure 18 for the two-class problem and Figure 19 for the seven-class problem.

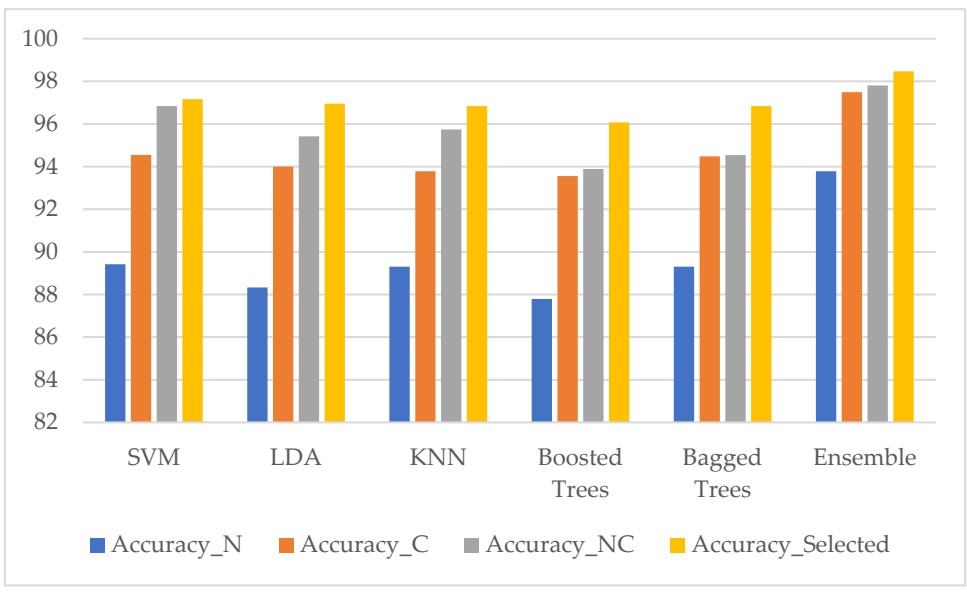

**Figure 18.** Classification performance of five classifiers and our ensemble classifier in terms of accuracy using four datasets (nuclei features only (Accuracy_N), cytoplasm features only (Accuracy_C), combining nuclei and cytoplasm features without features selection method (Accuracy_NC), and with features selection method (Accuracy_Selected)). (Herlev dataset for the two-class problem).

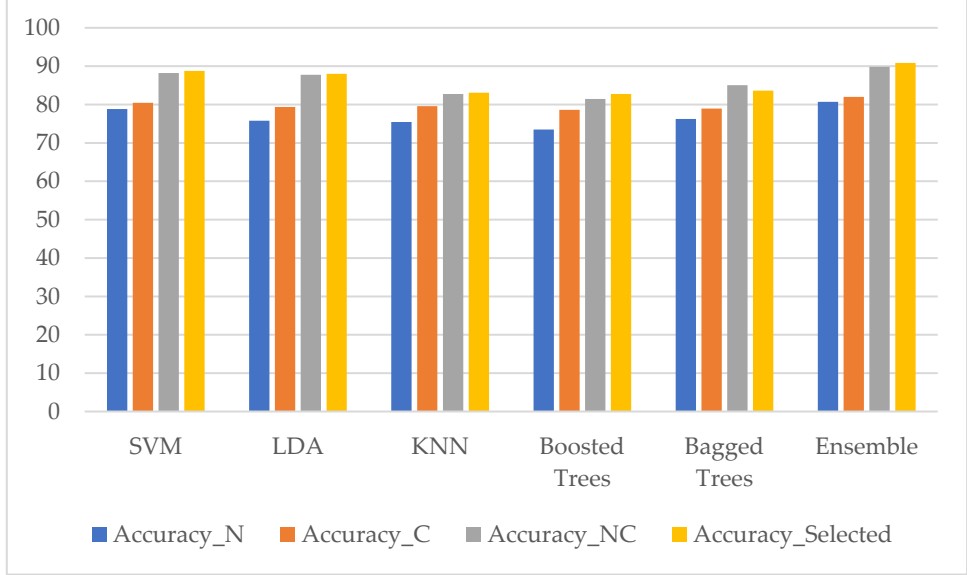

**Figure 19.** Classification performance of five classifiers and our ensemble classifier in terms of accuracy using four datasets (nuclei features only (Accuracy_N), cytoplasm features only (Accuracy_C), combining nuclei and cytoplasm features without features selection method (Accuracy_NC), and with features selection method (Accuracy_Selected)). (Herlev dataset for the seven-class problem).

The five performance measures of each classifier using selected features for the Herlev dataset are shown in Figure 20 for the two-class problem and Figure 21 for the seven-class problem.

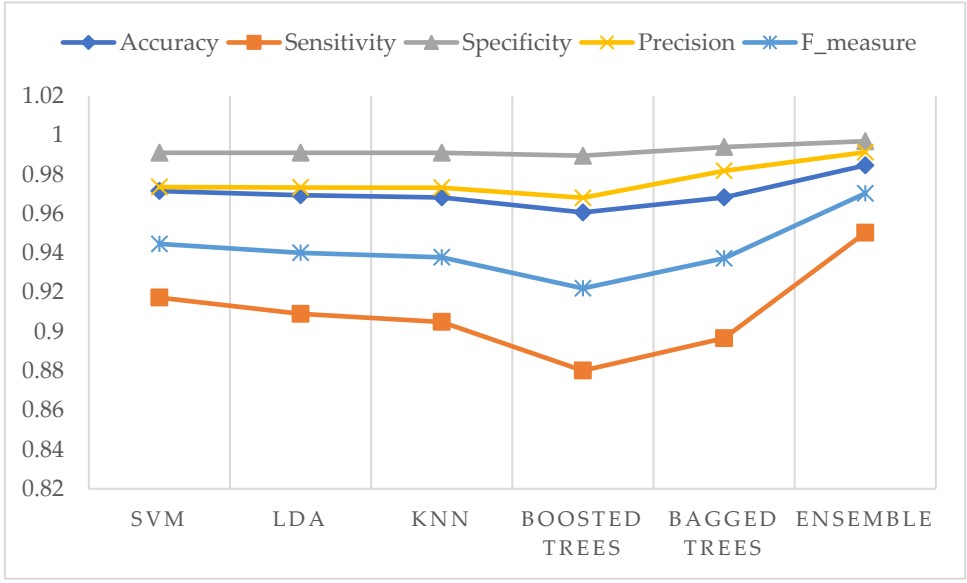

**Figure 20.** Evaluation of classifier's performance using selected features for the Herlev dataset in the two-class problem.

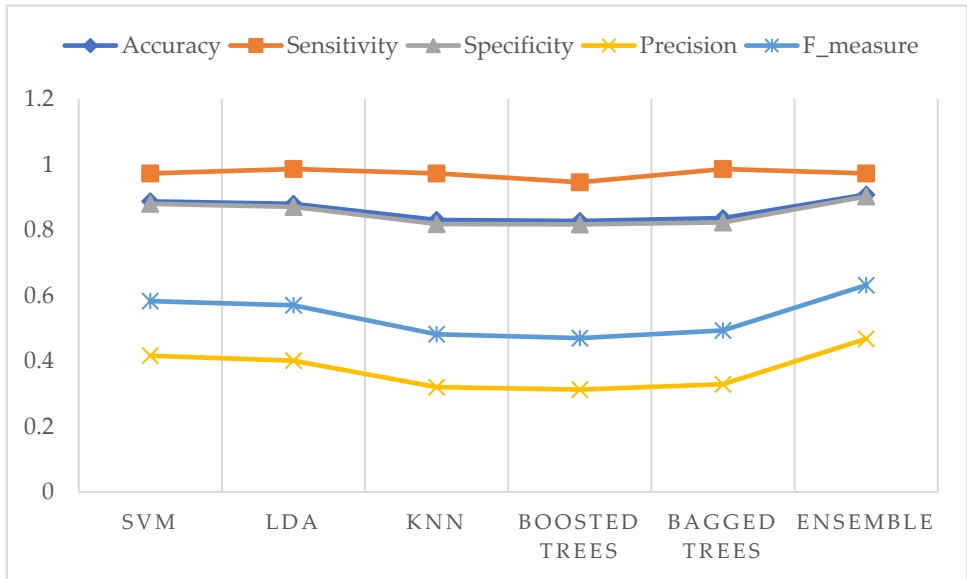

**Figure 21.** Evaluation of classifier's performance using selected features for the Herlev dataset in the seven-class problem.

## 4. Conclusions

This paper proposed a system for computer-assisted screening for cervical cancer using digital image processing of Pap smear images. Our proposed system consisted of six steps: image acquisition, image enhancement, cell segmentation, feature extraction, feature selection, and classification. Our proposed system first segmented each independent cell components, such as nucleus and cytoplasm, and then detected whether cells were cancerous or not through machine learning-based technique. There are several techniques, which had been proposed in the past in this direction. But the accuracy has not been found to be significantly accessible [40]. In our work, the average classification result showed an accuracy of 98.47% and 98.27% in the two-class problem and 90.84% (seven-class) and 94.09% (five-class) in multi-class problem using Herlev dataset and SIPaKMed dataset individually.

The main advantage of our proposed method is an increase in the predictive performance in separating the abnormal cells from the normal cells. The proposed system could be further enhanced by using other classifiers. Our proposed system showed better classification accuracy, sensitivity, and specificity than individual five classifiers. So, this framework could be used for cervical cancer screening system to detect women with precancerous lesions.

**Author Contributions:** K.P.W., Y.K., and K.H. designed and developed the study; K.P.W. performed the experiments, data analysis, validation, simulation, and manuscript writing; T.M.A. provided the medical reviews, annotated the pathologic cells, and verified the results; Y.K. and K.H. coordinated the whole efforts as the supervisors of this research. All authors have read and agreed to the published version of the manuscript.

**Funding:** This research received no external funding.

**Acknowledgments:** The authors highly appreciate the Asian University Network (AUN/SEED-Net) for their financial support to contribute to this research. Moreover, we also thank the Department of Pathology of Herlev University Hospital and the Department of Automation at the Technical University of Denmark for the Herlev dataset used in this study. We are also immensely grateful to the Department of Pathology, Faculty of Medicine, Kale Hospital, Myanmar, for greatly assisting the research.

**Conflicts of Interest:** The authors declare no conflict of interest.

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
