# Peer review of "Computer-Assisted Screening for Cervical Cancer Using Digital Image Processing of Pap Smear Images"

_applsci, doi:10.3390/app10051800_

Round 1

Reviewer 1 Report

This paper describes a machine learning approach to automate the reading of Pap smears.

Major comments are

the manuscript does not consider the "big picture". Why are pap smears of interest when there is HPV vaccination and many jurisdictions are moving to implement HPV testing in primary screening? I accept that not all jurisdictions doing this, and that it may be some time before these changes are implemented in the jurisdictions that have the resources to love in this way. When HPV testing is used in primary screening, Pap testing could be used in triage. However the interpretation of test could be influenced by knowledge of outcome of HPV test. How would that impact the automated screening? Moreover, in this situation, int is likely that the cells would be obtained from liquid based cytology samples rather than smears. Does that matter? In jurisdictions in which HPV vaccination has been implemented, the prevalence of cervical cytology abnormalities likely to be reduced. This has implications for positive predictive value (PPV) but also the reduced prevalence might have implications for the reliability of manual reading - that would be an argument in favour of automation It is stated that two datasets of images (one of single cells, one of multiple cells) were used to test the system. However, it is not clear whether the images related to samples that were obtained in primary screening (of asymptomatic women), follow -up of women who had an abnormality found in primary screening, or had been obtained during diagnostic workup of symptomatic women. In the abstract, but is stated that the automated method "has been evaluated against the results of manual tracing by experts" but if this is described in the main body of the manuscript, I have not been able to understand it. It would help the reader to understand the manuscript if many of the formula, figures and tables were moved to appendices I would strongly suggest that the authors include a clinician with expertise in cervical screening, and an epidemiologist or biostatistician, in their team

Reviewer 2 Report

The manuscript developed a novel automatic methods to analyze the pap smear results. It's very helpful for cervical cancer screening. The new methodology provide more accurate results. The manuscript is well-written and clearly. It's acceptable and publishable. 

Reviewer 3 Report

Cervical cancer is the fourth most frequent cancer in women with an estimated 570,000 new cases in 2018 representing 6.6% of all female cancers. Nearly all cases of cervical cancer can be attributed to infection with human papillomavirus (HPV). Over 99% of precancerous lesions (cervical dysplasia) and cervical carcinomas are caused by high-risk HPV infection. More than 200 strains of HPV have been identified, of which approximately 40 infect the anogenital region. 15−18 of these HPV strains have been classified as high-risk genotypes. Virtually all cervical neoplasias and cancers are attributable to high-risk HPV genotypes, and approximately 70% of all cervical cancer cases are attributable to types 16 and 18.

Almost all cervical cancers are either squamous cell carcinoma or adenocarcinoma. The major steps known to be necessary in cervical carcinogenesis include HPV infection, HPV persistence, progression to dysplasia, and invasion. Persistent infection with high-risk HPV types may lead to precursor lesions of the cervix, referred to as CIN, which is epithelial cellular change, where the ratio of the cell nucleus to the size of the cell is increased. CIN is graded as CIN1 (mild), CIN2 (moderate), or CIN3 (severe) depending on the proportion of the thickness of the epithelium showing mature, differentiated, and undifferentiated cells. CIN usually occurs in the transformation zone of the cervix near the squamocolumnar junction. Invasive cervical cancer develops from CIN – mild to moderate to severe CIN and then to cancer over a prolonged period of time, usually 7 to 20 years. Most mild CINs spontaneously regress, but some may progress to higher grade CIN. Moderate or severe CIN should be treated as it carries a much higher probability of progressing to invasive cancer, although a proportion of such lesions also regress or persist. It may then take a decade or more to develop invasive cervical cancer from CIN3.

Early detection, by screening all women in the target age group, followed by treatment of detected precancerous lesions can prevent the majority of cervical cancers. Currently, the standard treatment recommendations following diagnosis of CIN1 include monitoring for progression, whereas treatments for CIN2 and CIN3 include cryotherapy, thermoablation, loop electrosurgical excision procedure (LEEP) and cold knife conization (CKC). If women with CIN3 fail to receive treatment, then about 30% of them will progress to cervical cancer.

Cervical cancer screening was revolutionized in the early 1980s by the discovery of HPV as the single causative agents of the disease. In 1983, HPV type 16 (HPV16) was first identified in DNA from a biopsy sample of invasive cancer of the cervix, and in the following years, HPVs were reported as the main causative agents of cervical cancer. There is strong evidence that high-risk human papillomavirus (HR-HPV) DNA based screening is more sensitive for the detection of high-grade CIN (CIN2+) and is effective in prevention of ICC compared to cervical cytology and visual inspection. Internationally, transition to HPV primary screening is proceeding or planned in several countries, such as Mexico, Italy, the Netherlands, Australia, Sweden, and Scotland. European guidelines recommend HPV primary screening for organized, population-based screening.

In the US, HPV testing in combination with Pap cytology (co-testing) is recommended at five-year intervals for women between the ages of 30 and 65. The growing adoption of HPV-based screening can be explained by some advantages that HPV testing is expected to offer over cytology, such as higher sensitivity and reproducibility, the possibility to safely increase the time between screening visits, the potential for the screening process to be more efficient and cost-effective, and the opportunity to implement selfsampling to encourage screening participation in under- and never-screened populations, among others.

Win et al. have suggested an automated screening system for cervical cancer using digital images of pap smears. The screening system was based on four basic steps: cells segmentation, features extraction step, feature selection and classification. SIPaKMeD and Herlev datasets were used to prove the effectiveness of the proposed screening system.

Comments

1. The three authors are from Faculty of Engineering, King Mongkut’s Institute of Technology Ladkrabang, Bangkok and School of Information and Telecommunication Engineering, Tokai University, Tokyo. I do not think any of them have any medical background or really understand the concept of cervical cancer prevention. They use a data set of digital pictures of Pap smears from Herlev Hospital in Denmark. I suggest they include a pathologist at the Department of Pathology at Herlev Hospital as a co-author of the manuscript.

2. I do not think any of the authors are native speaking English. The manuscript should be proofread by native English speaking academic researchers. Especially the introduction of the manuscript is terrible.

3. Cervical cancer screening is actually not about detection of women with cervical cancer, but to detect precancerous lesions (high grade cervical intraepithelial neoplasia) that can be treated to prevent development of invasive cervical cancer. In the manuscript they use the term “cancer cells”, but abnormal cells in a precancer is not cancer cells, but dysplastic cells. Dysplasia is a cellular change which occurs to develop neoplasm.

4. In the description of five classes cells from SIPaKMeD (Multi-cells) dataset they use the terms “Dyskeratotic: moderate dysplastic cells” and “Koilocytotic: severe dysplastic cells”. First of all they should use the Bethesda classification system for cervical cytology (Normal, ASC-US, LSIL, ASC-H and HSIL). Second, a koilocyte is a squamous epithelial cell that has undergone a number of structural changes, which occur as a result of infection of the cell by human papillomavirus. Koilocytic cells are classified as low grade changes (LSIL), not “severe dysplastic cells”.

5. The accuracy of digital analysis of images of Pap smears should not be on a single cell level, but to classify the whole Pap smear into Normal, ASC-US, LSIL, ASC-H and HSIL.

6. The true sensitivity and specificity of cervical cytology is not the cytology diagnosis itself, but the ability to detect women with high grade lesions (CIN2+) using histology as a gold standard.

7. Usually, using machine learning, you need a training set and a validation set. You need to know the cytological diagnosis using the Bethesda system, and you need follow-up data with cervical biopsies.

8. There are 12 tables and 35 figures in the manuscript. I think most of them are not very interesting and could be moved to Supplemental. There should be enough with 2-3 tables and 4-5 figures in the manuscript.

9. Most of the formulas on page 14-20 could be moved to Supplemental.

10. I do not think automated screening systems for digital analysis of images of Pap smears is the solution of cervical cancer screening, but HPV primary screening.

Specific revisions

Page 1, line 11, “Cervical cancer can be prevented if precancerous changes are detected in an early stage” => “Cervical cancer can be prevented by having regular screenings to find any precancers and treat them”

(there is no early stage or late stage of precancerous lesions)

Page 1, line 12, “A pap smear is a simple, quick and painless screening procedure for cervical cancer” => “The Pap test looks for any abnormal or precancerous changes in the cells on the cervix”

(A Pap test is not a test for cancer, but a test to detect precancerous lesions that can be treated to prevent cervical cancer development)

Page 1, line 12-13, “However, the manual analysis of the pap smear is error prone due to human mistake” => “However, the manual screening of Pap smear in the microscope is subjective with poorly reproducible criteria”

Page 1, line 32, “Cervical cancer is the abnormal cell growth in the cervix” => “Cervical cancer is a cancer arising from the cervix. It is due to the abnormal growth of cells that have the ability to invade or spread to other parts of the body”

Page 1, line 32-33, “The main factor that may cause this cancer is infection with Human Papilloma Virus (HPV)” => “The most important risk factor for cervical cancer is infection with human papillomavirus (HPV)”

Page 1, line 33-34, “Early detection and confirmation of this kind of cancer is treatable and preventable” => “The goal of cervical screening is to identify and remove significant precancerous lesions in addition to preventing mortality from invasive cancer”

Page 1, line 34-35, “It is the second most leading causes cancers among the female all over the world. Over 27,000 women died each year due to cervical cancer. Approximately 80% of all cervical deaths occur in the developing country” => “Cervical cancer is the fourth most frequent cancer in women with an estimated 570,000 new cases in 2018 representing 6.6% of all female cancers. Approximately 90% of deaths from cervical cancer occurred in low- and middle-income countries.”

https://www.who.int/cancer/prevention/diagnosis-screening/cervical-cancer/en/

Page 1, line 37-39, delete “The infection with HPV virus is the main factor for cervical cancer. Other causes are cigarettes smoking, infection with HIV (human immunodeficiency virus), taking contraceptive pills for many years and having given birth more than three children”

Page 1, line 39-40, “This type of cancer cannot not notice at precancerous stages” => “Precancerous changes in the cervix usually don't cause any signs or symptoms”

Page 1, line 40-41, “When the cancer is progress, the most common signs are vaginal bleeding, discharge from the vagina and pain in sexual period” => “Symptoms of cervical cancer tend may include: irregular, intermenstrual (between periods) or abnormal vaginal bleeding after sexual intercourse; back, leg or pelvic pain; fatigue, weight loss, loss of appetite; vaginal discomfort or odourous discharge; and a single swollen leg. More severe symptoms may arise at advanced stages.”

https://www.who.int/cancer/prevention/diagnosis-screening/cervical-cancer/en/

Page 2, delete Figure 1 (Cervical cancer develops from the transformation zone of the cervix and is usually visible for the gynecologists. The tumor in Figure 1 is not in connection with the transformation zone and can not be seen by a gynecologist)

Page 2, line 44-60, delete “There are three main zones in cervix as shown in Figure 1. These three zones are endocervix, ectocervix and transformation or junction zone. Cervical cancers start at the junction zone. After infection with HPV, normal cells change into pre-cancerous stages within one to ten years. For invasion stage, it takes about thirty years. Mortality rates can be less if it is detected in early stage [3]. There are four stages in caner stage and these stages are divided upon the locations of its spreading. At first stage, cancer cells spread to the upper part of the cervix and uterus. At stage two, cancer cells grow over nearby tissues. At stage three, cancer cells spread to the lower part of the vagina and cause problem in urine flow. At final stage, cancer cells spread to other parts of the body[4]. Pap smear tests are used for examining cervical cells. In pap smear analysis, the segmentation process is also important. The main problem in cells segmentation is the segmentation of overlapping cells [5].

Pap test also known as cytology-based screening method are the major approach in clinical test. In current clinical tests, the examination of pap smear images is done by manual. The main problems are labor-intensive, long processing time and highly depend on expert cytologists. Therefore, automated screening systems are needed to solve these problems. In automated screening system, the segmentation of cervical cells is an essential task. Because, the features of are selected from each cell. The features of cells are very important to distinguish from normal to abnormal cells[6]”

Page 2, line 60-61, “My previous research work is emphasized on the classification of cervical cancer from single cell images”

You are three authors. Who am I?

Page 5, Table 1, “4. Mind dysplastic cells” => “ 4. Mild dysplastic cells”

Do you mean Bethesda ASC-US or LSIL?

Page 5, Table 1, “5. Moderate dysplastic cells”

Do you mean Bethesda ASC-H?

Page 5, Table 1, “6.Severe dysplastic cells” and “7.Carcinoma in situ”

I think both of them are Bethesda HSIL.

Page 5, Table 2, “Begin Cells” => “Benign Cells”

Page 5, Table 2, “3.Metaplastic: mild dysplastic cells”

Metaplastic cells are benign cells, not “mild dysplastic cells”

Page 5, Table 2, “4.Dyskeratotic: moderate dysplastic cells”

Do you mean Bethesda ASC-H?

Page 5, Table 2, “5.Koilocytotic: severe dysplastic cells”

This is confusing. Koilocytic cells are Bethesda LSIL, but “severe dysplasic cells” are Bethesda HSIL. What do you mean? LSIL or HSIL?

Page 18, line 400-401, “The main objective of the classification model is to classify the cancer cells with high precision and accuracy” => “The main objective of the classification model is to classify the abnormal cells with high precision and accuracy”

Page 30, line 565, “Our proposed system shows better classification accuracy than individual five classifiers”

What was the sensitivity and specificity of the classification system for detecting women with histological confirmed high grade lesions (CIN2+)?

Page 30, line 565-566, “So, this framework can be used for cervical cancer screening system to detect cancers at the early stage” => “So, this framework can be used for cervical cancer screening system to detect women with precancerous lesions”

(Screening with Pap smears is not to detect cervical cancer, but precancerous lesions that can be treated to prevent development of invasive cervical cancer)

References

Sorbye SW, Suhrke P, Reva BW, Berland J, Maurseth RJ, Al-Shibli K. Accuracy of cervical cytology: comparison of diagnoses of 100 Pap smears read by four pathologists at three hospitals in Norway. BMC Clin Pathol. 2017 Aug 29;17:18. doi: 10.1186/s12907-017-0058-8.

https://www.ncbi.nlm.nih.gov/pubmed/28860942

Round 2

Reviewer 1 Report

I thank the authors for their responses to my earlier comments, and note the addition of a pathologist as a co-author.

I accept your points about inability to access data from local hospitals in Myanmar. However, I still do not have an understanding as to the setting in which you would expect your system to be used.

It seems that you would anticipate that it would be used in low- and middle-income countries. Few of these have organized population-based programs. Some have opportunistic programs. I think these are mainly based on cervical smears, but in some there may be localized programs using primary HPV testing with triage by cervical smears, in others localized programs with primary screening by smears, and possibly some localized programs that use liquid based cytology in primary screening. 

I note that you have used a dataset obtained in Denmark. I was not able to find your ref 35, but found the following

Jantzen, J., Norup, J., Dounias, G., & Bjerregaard, B. (2005). Pap-smear Benchmark Data For Pattern Classification. In Proc. NiSIS 2005: Nature inspired Smart Information Systems (NiSIS), EU co-ordination action (pp. 1-9). Albufeira, Portugal: NiSIS.   The title implies that the samples were indeed from pap smears, and indeed it would appear that liquid based cytology was not introduced into Denmark until about 2009 - see https://www.tandfonline.com/doi/full/10.1080/0284186X.2017.1355110   I also note that you used a dataset developed in Greece. I cannot determine from ref 34 whether the samples were obtained in one of the organized cervical screening programs in Greece (https://www.ncbi.nlm.nih.gov/pubmed/11072209)   My concern is whether the characteristics of the images could have been affected by the circumstances in which the samples were collected. I note your assertion that your method can be applied to either Pap smear or liquid based cytology samples, but unless I have missed this, you do not say this in the revised manuscript, and you do not consider whether other factors could influence how your system performs.   In summary, I find the paper limited by the fact that the intended use of the system is not clear, and the nature of the specimens from which the images are derived is uncertain

Reviewer 3 Report

The revised manuscript has been improved, but there is still a lot of work to be done with the manuscript. I still think the manuscript has to been edited by an English-speaking native.

Some revisions:

Page 1, line 2-3, «Automated Screening System for Cervical Cancer Using Pap Smear Analysis» => «Computer Assisted Screening for Cervical Cancer Using Digital Image Processing of Liquid Based Cytology»

Page 1, line 4, «Thet Myo Aung 3» => «Thet Myo Aung 4»

Page 1, line 18-19, «the aim of this study is to develop an automated screening system that can prevent from the development of invasive cervical cancer» => “the aim of this study is to develop a computer assisted screening system for cervical cancer using digital image processing of liquid based cytology»

(I think the Herlev dataset is based on digital images of liquid based cytology (LBC) not conventional Pap smears. Conventional smears are often inadequate for digital imaging due to thick smear, which is not a problem of LBC due to even distribution of cells in the liquid medium).

Page 1, line 35-37, “Cancer is the uncontrolled growth of abnormal cells in the body and develops when the body's normal control system stops working. Old cells do not die and instead grow new cells out of control. These extra cells form a mass of tissue, called a tumor” => “Cancer is the uncontrolled growth of abnormal cells in the body. Rather than responding appropriately to the signals that control normal cell behavior, cancer cells grow and divide in an uncontrolled manner, invading normal tissues and organs and eventually spreading throughout the body.”

Page 1, line 38, delete “It is due to the abnormal growth of cells that can spread to other parts of the body”

Page 2, line 48-49, “Cervical cancer can be prevented by regular screening test and treated completely if precancerous changes are detected at early” => “Cervical cancer can be prevented by regular screening test if precancerous changes are detected and treated effectively, before cancer develops”

Page 2, line 49-51, “This kind of caner may not be known at precancerous stages and it takes long time to progress to cancer stage. Therefore, the earlier the detection, the greater the chances to diagnosis” => “Cervical cancer typically develops from precancerous changes over 10 to 20 years. The only way to know if there are abnormal cells in the cervix, which may develop into cervical cancer, is to have a cervical screening test”

Page 2, line 51, “Pap test is the common screening program for cervical cancer” => “Screening is testing of all women at risk of cervical cancer, most of whom will be without symptoms. A Pap test is commonly used to screen for cervical cancer”

Page 2, line 51-52, “This screening test is painless and comfort for women” => “A Pap smear is a simple, quick, and essentially painless screening test (procedure) for cancer or precancer of the uterine cervix”

Page 2, line 52-53, “It should be tested for women start from twelve years old until sixty years old” => “Cervical cancer testing should start at age 21. Women under age 21 should not be tested. Women between the ages of 21 and 65 should have a Pap test done every 3 years.”

https://www.cancer.org/healthy/find-cancer-early/cancer-screening-guidelines/american-cancer-society-guidelines-for-the-early-detection-of-cancer.html

Page 2, line 55-57, “But the weak point is that the manual analysis is a long time process and cannot give a reliable result because a pap smear slide image contains many thousands of cells. These cells need to be tested under a microscope by expert cytologists” => “The visual examination of the Pap smears is time consuming, very demanding, tedious, and expensive in terms of labour requirements. The cytotechnologists are laboratory professionals who study cells and cellular anomalies who go through a specialized training, typically of about one year”

Page 2, line 60-61, “The shape, size, texture and nucleus to cytoplasm ratio are the important features to distinguish the cancer cells into normal or abnormal” => “The shape, size, texture and nucleus to cytoplasm ratio are the important features to classify the cervical cells into normal and abnormal epithelial cells”

Page 2, line 73, “The studies of cervical cancer can be classified into the cell level classification” => “The studies of computer-assisted screening of cervical cytology can be classified into the cell level classification”

Page 3, line 109, “The system flow diagram of the proposed cervical cancer detection and classification system” => “The system flow diagram of the proposed computer-assisted screening of cervical cytology”

Page 5, line 140-141, “(4) Dyskeratotic: moderate dysplastic cells and (5) Koilocytotic:

141 severe dysplastic cells” => “(4) Dyskeratotic cells and (5) Koilocytotic cells”

Page 9, line 237, “The results of nuclei segmentation step are shown in Figure11” => “The results of nuclei segmentation step are shown in Figure 11”

Page 15, line 301-302, “class 4(dyskeratotic cells) and class 5(koilocytotic cells)” => “class 4 (dyskeratotic cells) and class 5 (koilocytotic cells)”

Page 20, line 367-368, “This paper proposed a system for automated screening system for cervical cancer detection using pap smear images” => “This paper proposed a system for computer-assisted screening for cervical cancer using digital image processing of liquid based cytology”
